# Precision Medicine for Chronic Endometritis: Computer-Aided Diagnosis Using Deep Learning Model

**DOI:** 10.3390/diagnostics13050936

**Published:** 2023-03-01

**Authors:** Masaya Mihara, Tadahiro Yasuo, Kotaro Kitaya

**Affiliations:** 1Infertility Center, Kouseikai Mihara Hospital/Katsura Mihara Clinic, Kyoto 615-8227, Japan; 2Department of Obstetrics and Gynecology, Otsu City Hospital, Otsu 520-0804, Japan

**Keywords:** chronic endometritis, computer-aided diagnosis, convolutional neural network, deep learning, fluid hysteroscopy

## Abstract

Chronic endometritis (CE) is a localized mucosal infectious and inflammatory disorder marked by infiltration of CD138(+) endometrial stromal plasmacytes (ESPC). CE is drawing interest in the field of reproductive medicine because of its association with female infertility of unknown etiology, endometriosis, repeated implantation failure, recurrent pregnancy loss, and multiple maternal/newborn complications. The diagnosis of CE has long relied on somewhat painful endometrial biopsy and histopathologic examinations combined with immunohistochemistry for CD138 (IHC-CD138). With IHC-CD138 only, CE may be potentially over-diagnosed by misidentification of endometrial epithelial cells, which constitutively express CD138, as ESPCs. Fluid hysteroscopy is emerging as an alternative, less-invasive diagnostic tool that can visualize the whole uterine cavity in real-time and enables the detection of several unique mucosal findings associated with CE. The biases in the hysteroscopic diagnosis of CE; however, are the inter-observer and intra-observer disagreements on the interpretation of the endoscopic findings. Additionally, due to the variances in the study designs and adopted diagnostic criteria, there exists some dissociation in the histopathologic and hysteroscopic diagnosis of CE among researchers. To address these questions, novel dual immunohistochemistry for CD138 and another plasmacyte marker multiple myeloma oncogene 1 are currently being tested. Furthermore, computer-aided diagnosis using a deep learning model is being developed for more accurate detection of ESPCs. These approaches have the potential to contribute to the reduction in human errors and biases, the improvement of the diagnostic performance of CE, and the establishment of unified diagnostic criteria and standardized clinical guidelines for the disease.

## 1. Introduction

Chronic endometritis (CE) is a localized infectious and inflammatory disorder of the uterine lining [1,2,3]. The major cause of CE is considered to be intrauterine infections by microorganisms frequently found in the urogenital areas, such as common bacteria (*Streptococcus* species, *Staphylococcus* species, *Escherichia* coli, and *Enterococcus* faecalis), *Mycobacterium* tuberculosis, *Mycoplasma* species, and *Ureaplasma* species. Antibiotic treatment against these microorganisms is effective to eradicate endometrial stromal plasmacytes (ESPCs), the landmark immunocompetent cells in this pathology. CE is drawing interest because of its association with female infertility of unknown etiology, endometriosis, repeated implantation failure following in vitro fertilization-embryo transfer cycles, and recurrent pregnancy loss as well as several obstetric complications (preeclampsia and preterm labor) and neonatal diseases in premature infants (periventricular leukomalacia and cerebral palsy) [4,5,6,7,8,9,10,11]. In contrast to acute endometritis, which manifests with intense symptoms such as pelvic pain, vaginal discharge, and systemic fever, CE is generally so asymptomatic or oligosymptomatic that it is often overlooked by affected patients and even by experienced gynecologists [2]. Given that the clinical course of CE, including onset, progress, and remission remains largely unknown, there are some controversies over the term “chronic” to describe this pathologic condition [3].

Currently, no universally accepted diagnostic criteria or clinical guidelines exist for CE. The diagnosis of CE has long relied on somewhat painful endometrial biopsy and histopathologic examinations combined with immunohistochemistry for CD138 (IHC-CD138), an ESPC marker also known as transmembrane heparan sulfate proteoglycan syndecan-1 [12]. With IHC-CD138 only, however, CE may be potentially over-diagnosed by misidentification of endometrial epithelial cells, which constitutively express CD138, as ESPCs [3]. Fluid hysteroscopy is emerging as an alternative, less-invasive diagnostic tool that can visualize the whole uterine cavity in real-time and enables the detection of several unique mucosal findings associated with CE. Meanwhile, the serious biases in the hysteroscopic diagnosis of CE are the inter-observer and intra-observer disagreements over the interpretation of the endoscopic findings [1]. Thus, each of these two diagnostic methods has its strengths and weaknesses. Additionally, in past studies, examiners set the definitions of CE according to their original standards or preferences, resulting in so many variances from one study to another. Some dissociation thereby exists in the histopathologic and hysteroscopic diagnosis of CE among researchers [1,2,3,13,14]. The optimization within individual diagnostic tools and integration between multiple diagnostic tools are craved for the establishment of standardized diagnostic criteria and/or unified clinical guidelines for CE.

One promising diagnostic tool for IHC is dual immunohistochemistry for CD138 and another potential ESPC marker multiple myeloma oncogene 1 (MUM-1), which are currently being tested. Artificial intelligence is being actively introduced into the medical field [15]. Deep learning is an artificial intelligence model that is intensively studied due to its well-fitting into medical image analysis. The advantage and value of the deep learning models in the situation of the clinical diagnosis are that the output can provide the probability of the disease, contrary to physicians being able to give only two judgments (presence or absence) [16]. Indeed, deep learning models have been shown to reduce errors and increase the accuracy and precision in the medical image diagnosis made by physicians [17]. Computer-aided diagnosis (CADx) using a deep learning model is being carried out for more accurate detection of ESPCs. These approaches have the potential that contributes to improving the diagnostic performance of CE and help us establish standardized diagnostic criteria and unified clinical guidelines for the disease. This article aimed to introduce some novel approaches for potential solutions to some questions on CE and give insights into precision medicine for women suffering from this disease.

## 2. Unsolved Questions on Histopathologic CE

As a localized mucosal inflammatory disease, the combination of endometrial biopsy and histopathology, along with immunohistochemistry, have been traditionally prioritized and emphasized as the diagnostic method of CE. The histopathologic features of CE contain superficial edema, increased stromal density, unsynchronized differentiation between endometrial epithelium and stroma, and infiltration of CD138(+) ESPCs. Of them, the most specific and sensitive finding of CE is the presence of multiple ESPCs [18,19,20,21,22]. These lymphoid cells are found in the endometrial stroma as scattered or clustered cells in CE. Antibiotic treatment is effective in the eradication of ESPCs in the affected infertile women, although it awaits further studies to determine if the histopathologic cure of CE improves the reproductive outcomes in the following infertility treatment cycles in these women [23,24,25,26,27,28].

Under a light microscope, typical plasmacytes are described as large-sized lymphocytes with a high nuclear-cytoplasmic ratio, basophilic cytoplasm, and eccentric nucleus with heterochromatin rearrangement (“spoke-wheel” or “clock-face” pattern). But several types of endometrial cells, including natural killer cells, macrophages, and stromal fibroblasts, present a morphological appearance that is close to ESPCs [18,19]. It is therefore very demanding and time-consuming to identify ESPCs by classical histopathologic examinations only. The introduction of IHC-CD138 significantly contributed to the development of the histopathologic diagnosis of CE with marked improvement of sensitivity (100% vs. 75%, IHC-CD138 vs. conventional tissue staining with hematoxylin/eosin or methyl green), specificity (100% vs. 65%, IHC-CD138 vs. conventional tissue staining), inter-observer variability (96% vs. 68%, IHC-CD138 vs. conventional tissue staining), and intra-observer variability (93% vs. 47%, IHC-CD138 vs. conventional tissue staining) [21,22]. Despite the application of IHC-CD138 and its spread in the clinical diagnosis of CE, there are some precautions for use.

First, CD138 is constitutively expressed not only in ESPCs but also in endometrial surface/glandular epithelial cells, especially on the basolateral sides. The currently available primary antibodies against CD138 also react to the epitope of CD138 both on ESPCs and endometrial epithelial cells. Although the immunoreactive intensity in endometrial epithelial cells is generally weaker than that in ESPCs, the conditions of the section preparation potentially cause the misidentification of endometrial epithelial cells as ESPCs, overestimation of ESPCs, and overdiagnosis of CE [3]. One of the potential risks of the over-diagnosis of CE is the overtreatment of infertile women using antibiotic agents. Multi-drug resistance is a serious global medical problem in the antibiotic treatment of infectious diseases. Antibiotic resistance is unexceptionally increasing in CE. In 2008, Cicinelli et al. [1] reported that <20% of CE was ineffective to single oral doxycycline treatment, whereas their update in 2015 demonstrated that multiple courses of antibiotic treatments failed to cure CE in 24.6% of the affected women, implicating the increment of multi-drug resistant CE [29]. In a retrospective/prospective survey from 2010 to 2020, we investigated the prevalence of antibiotic resistance in CE in a series of infertile women with a history of repeated implantation failure in three or more failed in vitro fertilization-embryo transfer cycles [30]. The prevalence of CE in the cohort did not change conspicuously for the ten years (30.2% from April 2010 to March 2015 vs. 31.7% from April 2015 to March 2020). The resistance (failed eradication of ESPCs) to the first-line 200 mg/day, 14-day oral doxycycline administration was observed in 21.2% of whole CE cases. Meanwhile, the prevalence of multi-drug resistant CE (regarded as the ineffectiveness of the first-line doxycycline and 14-day oral second-line metronidazole 500 mg/day and ciprofloxacin 400 mg/day treatment) in the whole CE cases was 8-fold higher in the latter five years (9.6% between April 2015 and March 2020) than in the earlier five years (1.3% between April 2010 and March 2015). In some women, we performed microbiome analysis in the vaginal and uterine cavities to seek the potential pathogens associated with multi-drug resistance in CE. However, any unique microbial genera/species and/or bacterial communities were not identified in the microbiota of their paired endometrial fluid and vaginal secretions [30,31]. These findings indicate the difficulty in the antibiogram designs and the choices of the third-line antibiotic agents for infertile women with multi-drug resistant CE.

Second, IHC-CD138 is not yet technically standardized for human endometrial tissue. The histopathologic diagnosis of CE may be influenced by multiple laboratory factors, such as the dilutions and incubation periods of the primary antibodies and the secondary detection systems, and the thickness, area, and number of the fields and/or sections evaluated for the detection of ESPCs. Indeed, it was shown that the choices for clones (B-B4 versus B-A38) and dilutions (1:1000 vs. 1:100 dilution of clone B-B4) of the primary antibodies brought about the differences in the diagnostic rates (more than 9% estimated difference) of histopathologic CE [5,32]. Moreover, the method and device of the endometrial biopsy can affect the diagnostic performance of histopathologic CE. ESPCs tend to form focal accumulation in the endometrial stromal compartments rather than homogeneous distribution. In addition, in some women with CE, ESPCs accumulate in the endometrial basal layer (the part that is not shed during menstruation) only [33]. Thus, ESPCs may be missed under observation within the small areas of the endometrial biopsy specimens [3]. IHC-CD138 using the “whole-wall” endometrial curettage may elevate the chance to detect ESPCs, but this method can pose serious harm to women desiring pregnancy, causing endometrial thinning and intrauterine adhesions/Asherman’s syndrome that can reduce their endometrial receptivity, along with some life-threatening gestational complications such as ectopic pregnancy and placenta accreta spectrum disorder [34,35,36,37].

## 3. CADx Using Deep Learning Model for Unsolved Questions on Histopathologic CE

While deep learning has been long utilized for the histopathologic diagnosis of the images of the specimens with conventional tissue staining, its application to IHC images is yet under development. Using a newly constructed framework, Zhang et al. [38] found that a deep learning model (entitled Global Scanner subnetwork) is capable of detecting CD138(+) ESPCs that are distributed sparsely in the endometrial stromal components effectively and efficiently as well as predicting the location map of CD138(+) ESPCs quickly in the whole slide images. In addition, they proposed a novel grid-oversampling strategy for the solution of the sample imbalance problems in preprocessing whole slide images.

MUM1 (also known as interferon regulatory factor 4) is a transcription factor expressed at the late plasmacyte-directed stages of B cell differentiation [39]. MUM1 is a promising ESPC marker that may be equivalent or possibly superior to CD138. Using a combination of IHC-CD138, IHC for MUM1 (IHC-MUM1), and conventional hematoxylin and eosin staining, Parks et al. [40] evaluated the presence of ESPCs in 311 endometrial biopsy specimens. While CD138 identified ESPCs in 15%–23%, MUM1 detected ESPCs in 48% of the samples with minor background staining, resulting in MUM1 as a more sensitive ESPC marker than in CD138. Meanwhile, Cicinelli et al. [41] showed that both the sensitivity and specificity in the detection of ESPCs were higher in IHC-CD138 than in IHC-MUM1, but IHC-MUM1 scored a higher inter-observer agreement value compared with IHC-CD138. The concomitant use of IHC-CD138 and IHC-MUM1 may thereby potentially make up for the shortcomings of each method in the histopathologic diagnosis of CE.

Recently, Jiang et al. [42] tested the feasibility of dual immunohistochemistry for these two plasmacyte markers for the detection of ESPCs. They found that CD138(+)/MUM1(+) ESPCs can be distinguished from false-positive cells that are immunoreactive to CD138 alone (14%) or MUM1 alone (24%). When the histopathologic CE was defined as 5 or more CD138(+)/MUM1(+) ESPCs per section, all of the sensitivity, specificity, and accuracy, of its diagnosis reached 100%. Interestingly, they further developed and trained an artificial intelligence system for the automatic identification of CD138(+)/MUM1(+) ESPCs and the diagnosis of histopathologic CE. To reduce the values of the false positive rates and improve the precision, they adopted the deep learning model that comprised the cascades of a high-performance anchor-free version of you-look-only-once (YOLOX), residual neural network-18 (ResNet18), and extreme gradient boosting (XGBoost) classifiers (Figure 1). The digitally photographed images of immunostained endometrial specimens were cropped using a sliding window. Following the discovery of the valid areas in the whole slide images using the Otsu algorithm [43], the images were entered into a trained YOLOX-s. A total of 2000 regions of interest in the whole slide images were chosen, and 1308 patches containing ESPCs were manually annotated with bounding boxes. After the mean precision value of 0.51 was yielded in the testing data set, the patches containing ESPCs and those not containing ESPCs were further processed for the training of the ResNet18 model as a post-processing module and XGBoost using the extracted shape, color, and texture features of ESPCs. Compared with the review results of the experienced pathologists, their deep learning model achieved sensitivity, specificity, and accuracy rates of 100%, 83.3%, and 91.4%, respectively, for the histopathologic diagnosis of CE.

XGBoost is an optimized distributed gradient-boosting library with a binary decision tree (yellow and blue dots) of algorithms that implement machine learning under the Gradient Boosting Decision Tree framework [44]. The residual from tree-1 is fed to tree-2 to decrease both the residual and onward. Distinct from Random Forest, each tree model in XGBoost minimizes the residual from its previous tree model. XGBoost performs the second-order Taylor expansion of the cost function and uses both the first and second derivatives.

ResNet is a deep residual neural network that replaces convolutional layers and pooling layers in a convolutional neural network (CNN) with fully connected layers [45]. It contains an input layer, an output layer, and in-between three sets of one dense block and two identity blocks with three hidden dense layers in both blocks. While the input is also connected to the output via another dense layer in a dense block, it is directly connected to the identity blocks.

YOLOX is a real-time object detection system characterized by the adoption of the decoupled head to improve the conflict between classification and regression tasks. For each level of the feature pyramid network, one 1 × 1 convolutional layer was adopted to reduce the feature channel to 256. Two parallel branches and 3 × 3 convolutional layers were then added for classification and regression tasks, respectively. The Intersection Over Union (IOU) branch was added to the regression branch [46].

These findings indicate the superiority of the dual IHC-CD138/MUM-1 to the single IHC-CD138 and the possibility of the clinical application of the deep learning model to the histopathologic diagnosis of CE. In addition, the development of deep learning models holds promise for the determination of the threshold/cut-off ESPC density to define histopathologic CE, which is another unsolved question in female infertility.

## 4. Unsolved Questions on Hysteroscopic CE

Fluid hysteroscopy is a handy and versatile diagnostic modality that can be performed in office gynecologic practice. Fluid hysteroscopy has been broadly utilized for real-time detection of female infertility-associated uterine cavity lesions, such as endometrial polyps, submucosal fibroids, intrauterine adhesions/Asherman’s syndrome, and uterine septum. Recent studies focus on the application of fluid hysteroscopy for endoscopic diagnosis of CE, instead of rather painful and hemorrhagic endometrial biopsy/suction and time-consuming histopathologic diagnosis/microbial analysis [13]. Thus, fluid hysteroscopy is expected to be a promising diagnostic tool that works both for infertile couples and reproductive endocrinologists if it can visually define and accurately identify the lesions with CE.

In 2019, based on the two rounds of the systematic review of the selected articles and the Delphi poll agreement, the diagnostic criteria for hysteroscopic CE were proposed by the International Working Group for Standardization of Chronic Endometritis Diagnosis as follows [47]:(1)strawberry aspect: localized/scattered large hyperemic endometrial areas flushed with white central points [48],(2)focal hyperemia,(3)hemorrhagic spots: focal lurid endometrium with sharp and irregular borders possibly in continuity with capillary,(4)micropolyposis: a cluster of typically less than 1 mm-sized protrusions on the focal or entire surface with a distinct connective vascular axis [49],(5)stromal edema: the thick and pale appearance of the endometrium in the proliferative phase originating from the stroma (a nonpathologic finding that is observed during the secretory phase).

Of these five hysteroscopic features, endometrial micropolyposis is the finding that endoscopists can easily visualize. Endometrial micropolyposis is attracting attention with the anticipation that suggesting the presence of histopathologic CE with the highest probability. In a retrospective study, Cicinelli et al. [49] first assessed the relationship between endometrial micropolyposis and histopathologic CE. While Endometrial micropolyposis was identified in a total of 11.7% of 820 women undergoing fluid hysteroscopy, histopathologic CE was detected in 93.7% of these women with endometrial micropolyposis. By contrast, histopathologic CE was less frequently identified in women without evident endometrial micropolyposis (10.8%), resulting in a very high prevalence of histopathologic CE in women with endometrial micropolyposis (odds ratio 124.2, confidence interval 50.3–205.4). In addition, endometrial micropolyposis is unexceptionally associated with other findings suggesting the presence of hysteroscopic CE, such as stromal edema and focal hyperemia. Meanwhile, endometrial micropolyposis was observed in 53.6% of women with histopathologic CE. The sensitivity, specificity, positive predictive values, negative predictive values, and diagnostic accuracy of the endometrial micropolyposis for the presence of histopathologic CE were 54%, 99%, 94%, 89%, and 90%, respectively. In another retrospective study that enrolled 910 women with a history of abnormal uterine bleeding, the same research group further demonstrated that the positive and negative predictive values rose to 98.4% and 94.5%, respectively, if fluid hysteroscopy detected the triad of endometrial micropolyposis, stromal edema, and focal (or diffuse) hyperemia [50].

Similarly, Zolghadri et al. [51] also reported the association between the findings of hysteroscopic CE and those of histopathologic CE, with a higher sensitivity (98.4%) and negative predictive values (97.82%), but the specificity (56.23%) and positive predictive values (63.5%) of the combination of the two hysteroscopic CE findings (endometrial micropolyposis and hyperemia) for histopathologic CE were lower. These results implicate that the likelihood of histopathologic CE is quite high if endometrial micropolyposis is identified in fluid hysteroscopy, whereas substantial proportions of women with histopathologic CE do not present the findings of endometrial micropolyposis. The limitation and biases of these studies [47,49,50,51] were that the diagnosis of histopathologic CE relied on the sole classical histomorphological findings including superficial stromal edema, increased stromal density, and pleomorphic stromal lymphocyte infiltrates, along with the detection of ESPCs based on conventional tissue staining using hematoxylin and eosin only.

Using more sensitive and specific IHC-CD138 based on the detection of CD138(+) ESPCs, the estimated diagnostic accuracy of endometrial micropolyposis (alone or in combination with other hysteroscopic findings) under the fluid hysteroscopy for prediction of the presence of histopathologic CE is calculated as 60–70% [52,53,54,55], although some variances are seen between the studies. In addition, the limitation and potential biases in these studies were the retrospective design. In a prospective study that enrolled 94 women with a history of repeated implantation failure or recurrent pregnancy loss, it was demonstrated that endometrial micropolyposis, stromal edema, and hyperemia often coexist within an individual and these three hysteroscopic features are predominant as the suggestive findings of histopathologic CE [54]. These findings were also confirmed in a meta-analysis [56,57].

Recently, Wang et al. [58] performed a unique study that included infertile women before proceeding to their first in vitro fertilization-embryo transfer treatment cycle. Both fluid hysteroscopy and endometrial biopsy/IHC-CD138 were performed in succession on the same day in the proliferative phase. The presence of one or more endoscopic findings of endometrial focal hyperemia/strawberry aspect, micropolyposis, and/or stromal edema was regarded as hysteroscopic CE, whereas the infiltration of ≥5 CD138(+) ESPCs in 1 HPF (= ≥50 CD138(+) ESPCs in 10 HPFs) was defined as histopathologic CE in this study. Serum concentrations were measured for pituitary hormones and ovarian steroids in the identical cycle. In infertile women with endometrial focal hyperemia/strawberry aspect, progesterone and basal follicle-stimulating hormone concentrations were significantly lower than in those with endometrial micropolyposis and/or stromal edema. On the contrary, in infertile women with endometrial focal hyperemia/strawberry aspect, body mass index and other serum markers (testosterone, anti-Mullerian hormone, and serum basal luteinizing hormone concentrations) were significantly higher than in those with endometrial micropolyposis and/or stromal edema. These findings indicate a close association between endometrial focal hyperemia/strawberry aspect and polycystic ovarian syndrome.

Additionally, endometrial focal hyperemia/strawberry aspect was seen more frequently in women with primary infertility than in those with endometrial micropolyposis and/or stromal edema, which were more prevalent in women with secondary infertility. These findings implicate that infertile women with a history of previous pregnancies are more likely to have endometrial micropolyposis and/or stromal edema. Thus, the presence of the conceptus in the uterine cavity may potentially increase the risk of local microbial infection. Moreover, the prevalence of histopathologic CE was much lower in the endometrial focal hyperemia/strawberry aspect group (10.1%) than in the endometrial micropolyposis group (63.2%) and endometrial stromal edema group (74.0%), indicating that the occurrence of histopathologic CE significantly differs depending on the features of hysteroscopic CE. In addition, the lower prevalence of histopathologic CE in the focal hyperemia/strawberry aspect group than in the other two groups implicates the low positive predictive value of endometrial focal hyperemia/strawberry aspect for histopathologic CE.

Furthermore, the researchers further compared the hysteroscopic CE findings before and after a 14-day, 200 mg/day oral doxycycline administration cycle. [58]. The total cure rate of hysteroscopic CE in the second-look fluid hysteroscopy performed in the proliferative phase in the next cycle (3–5 days after the cessation of the menstrual bleeding) was 75.82% (207/273). The cure rate of endometrial micropolyposis and stromal edema were 73.6% and 83.2%, respectively, but none in the endometrial focal hyperemia/strawberry aspect group were improved despite the antibiotic treatment. Collectively, judging by these findings, the endometrial focal hyperemia/strawberry aspect, the findings that are more prevalent in women with polycystic ovarian syndrome than in those with other infertility etiologies, may not represent the definitive endoscopic signs of CE.

Taken together, if endometrial micropolyposis is identified on fluid hysteroscopy, the likelihood of histopathologic CE is considerably high. Meanwhile, a substantial proportion of histopathologic CE does not present endometrial micropolyposis. If other features of hysteroscopic CE are accompanied by endometrial micropolyposis, the predictive accuracy for the histopathologic CE rises significantly [59]. On the other hand, the predictive value of classical endometrial polyps for histopathologic CE remains controversial. Further studies are essential to draw a conclusion regarding this question.

## 5. CADx Using Deep Learning Model for Unsolved Questions on Hysteroscopic CE

One of the clinical problems of the hysteroscopic diagnosis of CE is that the interpretation of the findings by gynecologists tends to be subjective [60]. The accuracy of the fluid hysteroscopic CE findings to predict histopathologic CE based on IHC-CD138 is moderate (accuracy 60–70%) as aforementioned. To minimize these kinds of human biases/errors, CADx systems using deep learning models are being actively introduced into gynecologic practice in parallel with other medical fields. For example, using a convolutional neural network (CNN) and visual attention mechanisms, Sun et al. [61] first developed a CADx system using a deep learning model, entitled HIENet, which can classify the photographic images of the hematoxylin and eosin-stained endometrial tissue preparations into four categories (normal endometrium, endometrial polyp, endometrial hyperplasia, and endometrial adenocarcinoma). Following the testing, training, and cross-validation, the model achieved an accuracy of 84.5% as well as an area under the curve value of 0.9829 with 77.97% sensitivity and 100% specificity. This deep learning model finally outperformed three expert pathologists and five existing CNN-based classifiers. Meanwhile, Zhang et al. [62] reported the establishment of the CNN-based deep learning model that is capable of the automatic classification of the endometrial lesions presented by hysteroscopic image inputs. The accuracy of the model in the five-category classification of endometrial lesions was 80.8 with a sensitivity and specificity of 84.0% and 92.5% for endometrial hyperplasia without atypia, 68.0% and 95.5% for atypical hyperplasia, 78.0% and 96.5% for endometrial cancer, 94.0% and 95.0% for endometrial polyp, and 80.0% and 96.5% for submucosal fibroids, respectively. In the task of classifying the lesions into two categories (benign or premalignant/malignant), the model achieved 90.8% accuracy, 83.0% sensitivity, and 96.0% specificity, respectively. Again, this model outperformed multiple experienced gynecologists. Thus, they concluded that the mode is able to support gynecologists to improve the overall accuracy of the diagnosis of endometrial lesions. Furthermore, Takahashi et al. [63] demonstrated a CNN-based automatic hysteroscopic image analysis model that distinguishes endometrial cancer lesions from normal endometrium. Compared with the standard diagnostic method (78.91–80.93%), the model employing the proposed continuity analysis (83.94–89.13%) and the combination of the three neural networks (90.29%) obtained higher accuracy, with the high scores of the corresponding sensitivity and specificity 91.66% and 89.36%, respectively.

Given the recent prominent progress in hysteroscopic image analysis, the CADx systems using deep learning models have the potential enough to be utilized for hysteroscopic prediction of the presence or absence of histopathologic CE. We launched a study to construct a CNN-based deep learning model for the prediction of histopathologic CE in infertile women using archival fluid hysteroscopic images. As the first step, we trained the model with Visual Geometry Group (VGG)-16 to detect endometrial micropolyposis (Figure 2). A portion of archival hysteroscopic images was randomly selected for testing and validation, whereas another portion was used for data augmentation and model training. The training accuracy and validation accuracy of the model gradually increased and reached a plateau over 105 and 90 epochs, respectively. The value for the area under the curve for diagnosing endometrial micropolyposis exceeded 0.90 (as of 6 November 2022) which was at a similar level with several experienced gynecologists (according to the comparison using the DeLong test [64]). As our model seems to be feasible for the detection of endometrial micropolyposis in hysteroscopic image analysis, we are proceeding to the next steps to test and train the model for the prediction of the cases without endometrial micropolyposis but with histopathologic CE, those with endometrial micropolyposis but without histopathologic CE, and those with other hysteroscopic CE findings and histopathologic CE.

VGG16 is developed by the Oxford Visual Geometry Group. VGG16 is composed of the convolution layers with a stride 1 and the same paddings and the max pooling layers with a stride 2, resulting in a total of 138 million parameters. One of its features is that all the convolutional kernels are of size 3 × 3 and max-pooling kernels are of size 2 × 2. This is in contrast to Alexnet, the winner of the ImageNet Large Scale Visual Recognition Challenge 2012, adopting several different sizes of kernels. The other is the implementation of the deeper sixteen layers, which doubles the eight layers of Alexnet. At the end of the stream of VGG, the data passes through the fully connected layers and rectified linear unit function [f(x) = x+ = max (0, x)], an activation function to separate specific excitation and unspecific inhibition. They were followed by the softmax function [(z)i = ezi/(ez1 + ez2 + ez3 + … + ezK) for i (= 1, 2, 3,…, K) and z (= z1, z2, z3,…, zK)] for the normalization of the output of the CNN to a probability distribution over predicted classes based upon Luce’s choice axiom [65]. The model repeatedly iterates to reduce the cross-entropy loss to enhance accuracy.

## 6. Conclusions

The negative impact of CE on reproductive outcomes in infertile women recently comes to light [65,66,67,68,69,70]. The etiology and pathogenesis of CE are becoming clear with the expansion and development of research incorporating cellular and molecular biological approaches. Meanwhile, the cause-effect relationship between CE and female fecundity yet remains to be determined [71]. Different studies so far adopted different definitions for CE. To get the right answers and solve the inconclusive questions on CE, the establishment of unified diagnostic criteria that integrated histopathology and hysteroscopy (along with microbiome analysis) is essential. To reach the goal, more studies with rigorous designs are indispensable to delineate the distinct boundaries between CE and non-CE cases, although the CADx systems using deep learning models [72,73,74] are expected to help us establish the diagnostic criteria and standardize the clinical guidelines for this yet elusive disease and provide precision medicine for infertile women.

## Figures and Tables

**Figure 1 diagnostics-13-00936-f001:**
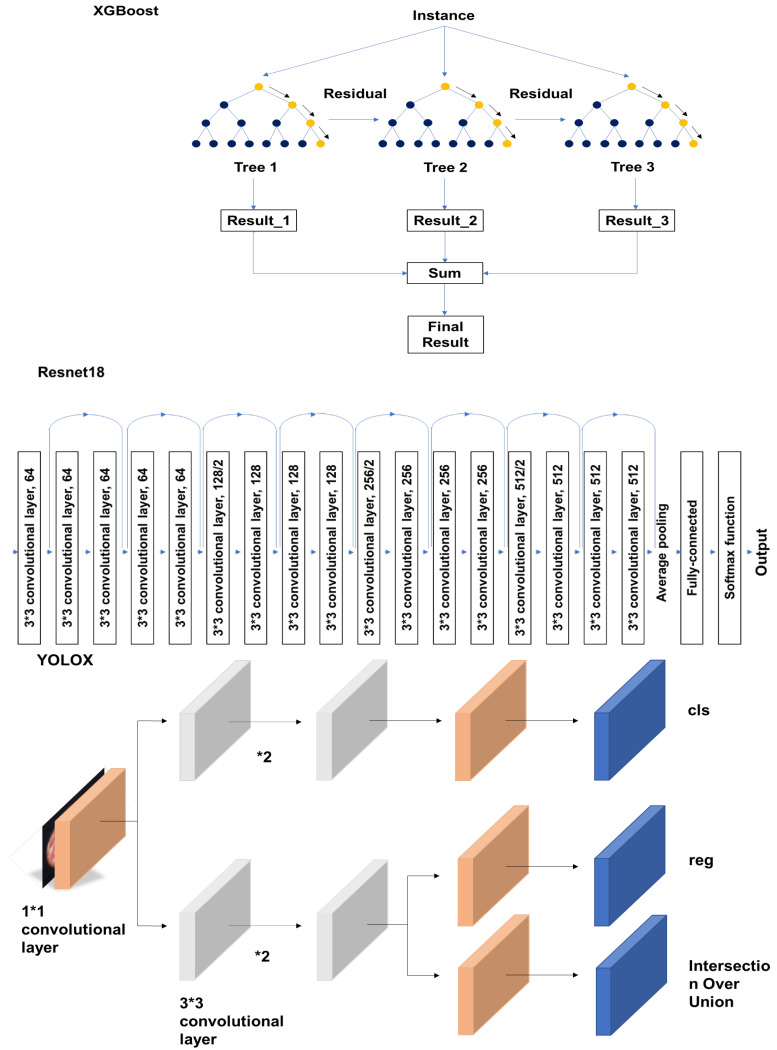
Schema of simplified architectures of XGBoost, Resnet18, and YOLOX.

**Figure 2 diagnostics-13-00936-f002:**
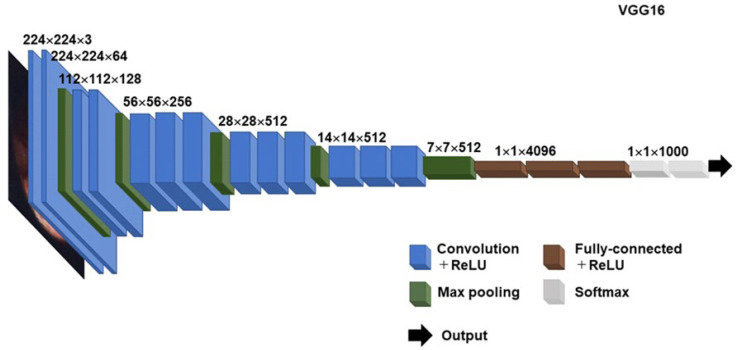
Basic architecture of VGG16 CNN model.

## Data Availability

Not applicable.

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
