# Peer review of "Precision Medicine for Chronic Endometritis: Computer-Aided Diagnosis Using Deep Learning Model"

_diagnostics, 2023, doi:10.3390/diagnostics13050936_

Round 1
Reviewer 1 Report
Chronic endometritis seems to be very important problem in infertility and reccurence abortions. I've red thist artice with great interest. In my opinion it is a veri valuable article adding new information about topic.
Author Response
Thank you very much for your effort and comments.
English was checked with a native speaker.

Reviewer 2 Report
DEAR AUTHORS,
I READ WITH GREAT INTEREST THE MANUSCRIPT, WHICH FALLS WITHIN THE AIM OF THIS JOURNAL. IN MY HONEST OPINION, THE TOPIC IS INTERESTING ENOUGH TO ATTRACT THE READERS’ ATTENTION. NEVERTHELESS, AUTHORS SHOULD CLARIFY SOME POINTS AND IMPROVE THE DISCUSSION, AS SUGGESTED BELOW. AUTHORS SHOULD CONSIDER THE FOLLOWING RECOMMENDATIONS:
IN MY OPINION YOU HAVE TO IMPROVE THE PAPER REFERING IN THE TEXT HOW ENDOMETRITIS COULD BE AN UNDIAGNOSTICATED CAUSE OF RECURRENT IMPLANTATION FAILURE (RIF) especially IN PTS THAT HAVE TRIED ALL THE KNOWN TECHNIQUE IN LITERATURE AS INJECTION OF EMBRYO CULTURE SUPERNATANT TO THE ENDOMETRIAL CAVITY IN IVF PROGRAMM.
ENDOMETRITIS CAN BE DETERMINATE AS WELL DY ENDOMETRIOSIS OR ADENOMYOSIS THAT CAN REDUCE THE POSSIBILITY TO GET PREGNANT .IN THIS CONTEXT LYVFESTILE AND DIET PALT AN IMPORTANT ROLE.
Atificial intelligence actually is really helpfull in the infertile pathway also in pts with chronic endometritis.
INJECTION OF EMBRYO CULTURE SUPERNATANT TO THE ENDOMETRIAL CAVITY DOES NOT AFFECT OUTCOMES IN IVF/ICSI OR OOCYTE DONATION CYCLES: A RANDOMIZED CLINICAL TRIAL.
THE FUTURE IS COMING: ARTIFICIAL INTELLIGENCE IN THE TREATMENT OF INFERTILITY COULD IMPROVE ASSISTED REPRODUCTION OUTCOMES—THE VALUE OF REGULATORY FRAMEWORKS.
CHRONIC ENDOMETRITIS AND ALTERED EMBRYO IMPLANTATION: A UNIFIED PATHOPHYSIOLOGICAL THEORY FROM A LITERATURE SYSTEMATIC REVIEW
IMPACT OF LIFESTYLE AND DIET ON ENDOMETRIOSIS: A FRESH LOOK TO A BUSY CORNER.
TRANSLATIONAL ANIMAL MODELS FOR ENDOMETRIOSIS RESEARCH: A LONG AND WINDY ROAD
Author Response
Thank you very much for your effort and comments.
We added these references (No.66-70) and discussion (Please find the revised parts with red letters/yellow background) to the manuscript.
